# The Comparison of Disordered Eating, Body Image, Sociocultural and Coach-Related Pressures in Athletes across Age Groups and Groups of Different Weight Sensitivity in Sports

**DOI:** 10.3390/nu15122724

**Published:** 2023-06-12

**Authors:** Migle Baceviciene, Rasa Jankauskiene, Renata Rutkauskaite

**Affiliations:** 1Department of Physical and Social Education, Lithuanian Sports University, 44221 Kaunas, Lithuania; renata.rutkauskaite@lsu.lt; 2Institute of Sport Science and Innovations, Lithuanian Sports University, 44221 Kaunas, Lithuania; rasa.jankauskiene@lsu.lt

**Keywords:** eating disorders, athletes, sociocultural and sport-related pressures, body appreciation

## Abstract

The aim of the present study was to compare disordered eating (DE), body image, and sociocultural and coach-related pressures between athletes of different age groups (adolescents and adults) and between athletes participating in weight-sensitive (WS) and less WS groups. A total of 1003 athletes participated in this study. The age range of the sample was 15 to 44 years, and the mean age was 18.9 ± 5.8 years (51.3% were female). Athletes who voluntarily agreed to participate in the study were provided with the study measures on DE, body image and sociocultural attitudes towards appearance. Vomiting, laxative misuse and excessive exercise were more prevalent in adolescent female athletes than adults, while dietary restraint was more common in adult male athletes than adolescents. Adolescent female athletes experienced higher sociocultural (family, peers) and sport-related (coach) pressures and a less positive body image compared with adult female athletes. Adult male athletes experienced higher overweight preoccupation, more DE and unhealthy eating habits, and engaged in more frequent self-weighing behaviour compared with adolescent males. When the effect of weight sensitivity in sports was tested, a higher prevalence of DE and overweight preoccupation, more frequent self-weighing, and higher body-image-related pressure from coaches were observed in female athletes participating in aesthetic weight-sensitive (WS) sports as compared with those participating in less WS sports. No differences in positive body image were observed in female WS and less WS sports. Special DE prevention and positive body image promotion programs are necessary for female competitive athletes and parents of adolescent female athletes, especially those participating in aesthetic ones. For adult male athletes, special programs aiming to promote healthy eating should be implemented to prevent DE and body image concerns. Special education about DE prevention is compulsory for coaches who train female athletes.

## 1. Introduction

### 1.1. The Prevalence and Consequences of Disordered Eating in Athletes

Disordered eating (DE) and eating disorders (ED) are one of the most common mental health illnesses in athletes [1,2,3]. DE can have devasting effects on athlete health and performance [4]. DE and ED are classified in WHO ICD-10 and in the fifth edition of the Diagnostic and Statistical Manual of Mental Disorders (DSM-V) as anorexia nervosa, bulimia nervosa, binge eating disorder and eating disorders not otherwise specified (EDNOS). EDNOS is hereafter termed DE. 

In athletes, DE might be understood in the continuum model starting with healthy dieting and exercise behaviours and the occasional use of more extreme weight loss methods, ranging to DE and ending with clinical EDs such as anorexia and bulimia nervosa [2,4,5,6]. Competitive athletes might move forward and back along the spectrum of the eating behaviour continuum at different points of their training cycle (i.e., during the preseason, competitive season, and off-season) and career [4]. DE is a set of disturbed eating patterns; unlike an ED, it does not conform to the clinical diagnosis of anorexia and bulimia nervosa or binge eating disorder. DE might manifest as unhealthy weight control behaviours, compensatory behaviours such as excessive exercising, binge eating, constant dieting, compulsive eating, fasting, self-induced vomiting, use of laxatives, skipped meals, use of weight loss supplements and other behaviours [2,4,5]. The prevalence of DE in competitive athletes is higher than that of ED [1,7]. Up to 45% of female athletes and up to 32.5% of male athletes are described as having DE or ED [1,8,9]. 

The consequences of DE attitudes and behaviours are related to the health and performance of competitive athletes [2,4,6]. DE in athletes is related to low energy availability (LEA) and the potential development of relative energy deficiency in sports (RED-S) [10]. LEA occurs when there is a mismatch between energy intake and exercise load, leaving insufficient energy to cover the body‘s other needs [2,11]. If DE overlaps with LEA, it negatively affects bone health, menstrual function, the endocrine, cardiovascular, gastrointestinal systems, and psychological functioning [4,11,12]. In adolescents, long-term consequences of LEA might be irreversible, since DE and LEA might have a negative effect on growth and maturity, reproductive functions, and bone mineral density [10,11]. DE also decreases athletes’ psychological functioning [1,4]. Psychological consequences include depression, anxiety, social isolation, substance use, self-harm, and increased risk for suicide [1]. Further, DE might increase the risks of injury and illnesses that compromise the training regime and its quality, training adaptation, and fatigue recovery [12,13]. The training might become interrupted and less effective, and impairments might affect the results of the competition [10,11].

The analysis of factors that are associated with the development and prevalence of DE in athletes is important for DE prevention practice in athletes. The main factors predisposing athletes to DE and ED are biological, psychological, sociocultural, and gender-based [2]. A significant part of the studies that previously analysed risk factors for eating psychopathology in athletes was implemented in samples of similar age and participating in one kind of sport (i.e., gymnastics, swimming) or in one group of sports (i.e., combat sports) [14,15,16]. To have a deeper understanding of DE and its associated factors, it is necessary to assess them in large samples of athletes of different ages participating in various sports [14]. In the present study, we aimed to expand scientific knowledge by comparing DE, body image, and sociocultural factors in different age groups of competitive athletes and comparing athletes participating in groups of different weight sensitivity in sports. Further, most of the studies concerning DE in athletes were implemented using samples of Scandinavian and North American collegiate athletes [10,12,17]. To our best knowledge, there are no studies assessing DE and factors associated with the development of DE in a sample of athletes from Eastern European countries; therefore, one of the objectives of the present study is to provide data on this topic. 

### 1.2. Disordered Eating, Sociocultural and Coach-Related Appearance Pressures, and Body Image in Athletes: The Effect of Age and Weight Sensitivity in Sports

The effect of age on DE, sociocultural attitudes towards appearance (general and sports-related), and body image are less studied. Research on adolescent athletes is scarce, and sport-specific risk factors for DE are less studied. Adolescence is a transition period from childhood to adulthood and a period of significant psychophysiological changes that require proper nutrition [18]. In this period, adolescents usually start specialising in a particular sport and experience major bodily changes that do not usually meet sociocultural and/or sport-related expectations. Therefore, they might experience higher general and coach-related pressures compared to adult athletes. In adolescence, DE and ED most frequently occur [19,20]. A recent systematic review concluded that age did not moderate DE in women athletes [21]. However, the effect of age on DE in male athletes is less studied. Thus, in the present study, we aimed to extend knowledge on this issue. 

Among the sport-specific factors, the most important factor associated with DE and ED in athletes is participation in weight-sensitive (WS) sports [6,8]. In WS sports, leanness and lower body weight play a major role in terms of performance, and lower body weight gives an athlete an advantage [22]. DE is more prevalent in sports with weight classes (i.e., rowing), aesthetic sports (i.e., figure skating, gymnastics), and sports where having a lower body weight is seen as advantageous (i.e., cycling) [4,8,23,24,25]. It is estimated that more than half of athletes participating in WS sports demonstrate DE behaviours, practising rapid weight loss (RWL) during competitive periods [16], and adolescent girls participating in aesthetic sports have significantly lower body fat compared with their peers participating in other sports [26]. Less is known about the differences in general sociocultural and coach-related pressures comparing athletes of different ages and genders participating in WS and less WS sports. Therefore, in the present study, we aimed to extend the current knowledge on this topic.

Studies show that athletes demonstrate less negative and more positive body image compared with non-athletes [21,27,28,29,30]. In male athletes, negative body image is not associated with DE [8]. However, in female athletes, dissatisfaction with body image is one of the strongest predictors of DE and ED [31]. Findings comparing the negative body image of WS and less WS sports are contradicting. Some studies found that females who participate in aesthetic sports experience a more negative body image [32]; however, others demonstrate the opposite [33,34]. Therefore, it is important to continue research on this topic. Finally, the majority of studies were focused on negative body image, and positive body image is understudied in competitive athletes [30]. Positive body image is associated with body functionality appreciation (appreciation for what one’s body can do), self-esteem, self-compassion, and psychological well-being, and it is protective for DE in women of general populations [35,36]. Findings of the previous studies suggested that older women report appreciating their bodies more readily than younger women [37]. To inform DE prevention programs in athletes, it is important to understand the differences in positive body image in various age groups of athletes and in groups of weight sensitivity in sports controlling for gender. In the present study, we aimed to assess if positive body image (operating as body appreciation) is more prevalent in adult athletes compared to adolescents and in less WS sports participating athletes compared to those participating in WS sports. 

The sociocultural model of Petrie and Greenleaf [38,39] posits that sociocultural (media, family, peers) and sport-specific pressures lead to DE through the internalization of appearance ideals, body dissatisfaction, and dietary restraint in athletes. Athletes are vulnerable to general pressures to attain stereotyped body ideals and also experience sport-specific pressures from sports environments (coaches, teammates, sports uniforms) [40]. More than 60% of elite athletes from leanness-focused and non-leanness-focused sports reported pressure from coaches concerning body shape [32]. Pressures from a coach are considered one of the main risk factors for body image concerns and DE in athletes [24,41,42]. However, less is known about the prevalence of sociocultural and sport-related pressures and internalization of appearance ideals in athletes of different ages. It is unclear if younger athletes experience more sociocultural and coach-related pressures compared to adult athletes. Adolescents spend a significant part of their time on social networking, which is a major source of body image concerns [43]; therefore, adolescent athletes might experience higher general pressures to attain stereotyped body ideals and to internalize these ideals more compared to adult athletes. Furthermore, adolescent athletes might perceive higher appearance pressures from coaches compared to adults. Finally, it is unclear if athletes participating in WS sports perceive higher sociocultural and coach-related pressures and internalize stereotyped appearance ideals to a greater extent compared to those participating in less WS sports. However, there is lack of empirical studies addressing these questions. Knowledge on these issues is important for science and evidence-based practice preventing DE in athletes [14]. 

### 1.3. The Present Study

The aim of the present study was to compare DE, body image and sociocultural and coach-related pressures between athletes of different age groups (adolescents and adults) and between athletes participating in WS and less WS groups. In the present study, we hypothesized that DE, body image concerns, and general sociocultural and coach-related pressures towards appearance would be more prevalent in adolescent athletes compared with adults controlling for gender. Next, we expected that DE, body image concerns, and sociocultural pressures, including pressures from coaches, would be higher in WS sports compared to less WS sports, controlling for age and gender. Finally, we expected that positive body image would be higher in adults compared to adolescents. We developed no assumptions for positive body image differences in groups of different weight sensitivity.

## 2. Materials and Methods

### 2.1. Procedure

This cross-sectional study was approved by the Lithuanian Sports University Social Research Ethics Committee (Protocol No. SMTEK-37, 27 May 2021). The online survey was implemented by using the Survey Monkey platform from January to June of 2022. Lithuanian athletes from different sports, who had competed for at least two years in professional competitive sports and were formally included in the lists as members of sport schools and/or sport clubs or belonged to national sports teams, were invited to participate in the survey. Persons who were not involved in competitive sports were excluded from the study. Information about the organized survey was spread via emails and sent to the following organizations: National Sports Federations, sport schools or gymnasiums, and sport centres across the country. Additionally, for better survey responses from respondents, members of national teams were contacted personally via social networks. Most recruited schools and organizations agreed to disseminate the online questionnaire with the help of coaches. The email contained an invitation to participate in the study, a participant information form, and a link to the online survey. The relevant sporting body emailed this information to athletes aged over 18 years and to the parents or guardians of athletes aged <18 years with a request to forward the information and survey link to the athlete. 

The inclusive criteria were being an athlete that competed for at least two years in any competitive sport (participating in sports competitions of one sport) and being an official member of a sports school, sport club, or national sports team. Exclusion criteria were not competing in any sport, competing in any sport for less than two years, not being officially listed as a member of a sports school, sport club, or national team.

The survey was completed anonymously. Before starting the survey, participants were informed that the study aimed to investigate lifestyle and health-related habits in competitive athletes, the time required to fill in the form (about 30 min), and that no identifying information will be collected. After the formalization, all the participants were asked to provide consent to participate by ticking one option “I agree to participate” or “I disagree to participate”. Those who disagreed were acknowledged and the survey was terminated, while those who agreed were provided with the study measures. Participation was voluntary, and only the researchers had access to the data. 

According to the Lithuanian sport centre, in 2021, 140,240 athletes participated in high-skill sports competitions, and 39,567 of them were female. Based on an a priori power calculation (small-to-medium effect size, α = 0.05, 95% power, *n* = 384), a minimum of about 400 women and 400 men would be recruited.

### 2.2. Participants 

One thousand three athletes (*n* = 1003) participated in this study. Of these, 488 (48.7%) were males (mean age 18.8 ± 5.7 years), and 515 (51.3%) were females (mean age 19.0 ± 5.9) from age 15 to 44 years. The mean age in both gender groups was similar: 18.8 ± 5.7 years in males and 19.0 ± 5.9 years in females. In total, 56% of all study participants were <18 years, while 44% were ≥18 years. The duration of involvement in competitive sports varied from 2 to 11 years (mean ± SD: 7.0 ± 3.2 years). Athletes reported from 2 to 21 h (mean ± SD: 11.2 ± 5.0) of exercise per week: of these, 10.8% exercised 2–5 h/week, 31.3% 6–9, 37.0% 10–15 and 20.9% 16–21 h/week.

### 2.3. Study Measures

The body mass index (BMI) was calculated by using self-reported weight (kg) and height (cm): weight (kg)/height (m^2^). The BMI ranged from 15.2 to 37.0 kg/m^2^ (mean ± SD: 21.6 ± 2.9 kg/m^2^). The mean BMI in gender groups was as follows: in males 22.2 ± 3.2 kg/m^2^, in females 21.0 ± 2.5 kg/m^2^. Using the criteria recommended by the WHO, all the athletes were classified into underweight (4.4%), normal weight (85%) and overweight/obesity (10.6%) groups. For athletes of age <18 years, the International Obesity Task Force (IOTF) cut-offs were applied [44].

According to the proposed classification of sports [22,45], all the athletes were classified into WS aesthetic sports (for example, gymnastics, competitive dancing, figure skating, body building), other WS sports (endurance/gravitational, for example, swimming, athletics, cycling and weight class, for example, boxing, martial arts) and less WS sports (ball games, for example, basketball, handball and technical sports, for example, fencing, shooting). In the sample of athletes, 6.4% of males and 20.4% of females participated in WS aesthetic sports (*n* = 136), 48.2% and 32.0% in other WS sports (*n* = 400), and 45.5 and 47.6% in less WS sports (*n* = 467), respectively.

The Lithuanian version of the Eating Disorder Examination Questionnaire 6 (EDE-Q 6) was used to examine disordered eating (DE) behaviours in athletes [46]. The questionnaire consists of 28 questions and statements with the response options ranging from 0 (never) up to 6 (always). Statements 1–12 and 19–28 were used to calculate the final score, averaging the response options, where a higher score indicates more frequent DE behaviour and greater body weight and shape concerns. For this study, we used only a total score. Next, we used items 2, 14, 16, 17, and 18 to define any occurrence of DE during the last 28 days [47]. Dietary restraint was a behaviour described as going for “long periods of time (>8 h) without eating anything at all in order to influence your shape or weight” (EDE-Q 6 item 2); binge eating distinguished by loss of control (or objective binge eating) was an episode described by eating a large amount of food with the feeling of losing self-control during consumption (EDE-Q 6 item 14); self-induced vomiting was an episode described as making “yourself vomit as a means of controlling your shape or weight” (EDE-Q 6 item 16); laxative misuse was an episode described as going “to take laxatives as a means of controlling your shape or weight” (EDE-Q 6 item 17); and excessive exercising was an episode described as exercising vigorously in “a driven or compulsive way as a means of controlling your weight, shape or amount of fat, or to burn off calories” (EDE-Q 6 item 18). Psychometric characteristics of the Lithuanian version of the EDE-Q 6 were tested in our previous research [48]. For this study, Cronbach’s α was 0.92 in females and 0.88 in males.

To assess negative body image, two subscales, the Overweight Preoccupation subscale and the Self-Classified Weight subscale from the Lithuanian version of the Multidimensional Self-Relations Questionnaire Appearance Scales (MBSRQ-AS), were used [49]. The Overweight Preoccupation subscale (4 items) assesses a construct reflecting fat anxiety, weight vigilance, dieting, and eating restraint, while the Self-Classified Weight subscale (2 items) reflects how one perceives and labels one’s own weight, from very underweight to very overweight, irrespective of actual body fat mass. Each subscale was rated on a five-point Likert scale. The final score was calculated by averaging the response options. Psychometric properties of the Lithuanian translation of the MBSRQ-AS were tested previously in an adult sample [50]. In this study, for the Overweight Preoccupation subscale, Cronbach’s α was 0.80 in females and 0.68 in males, while for the Self-Classified Weight subscale, it was 0.77 in females and 0.67 in males.

The Lithuanian version of the Body Appreciation Scale 2 (BAS-2) was employed to assess positive body image [51]. A unidimensional scale reflects a positive attitude towards one’s own body and the ability to resist body beauty standards. The BAS-2 contains 10 statements with a 5-point Likert scale from never (1) to always (5). A final score is calculated by averaging the response options to demonstrate greater body appreciation. A unidimensional factor structure and good psychometric properties of the Lithuanian translation of the BAS-2 were demonstrated in our previous study [52]. In this study, for the BAS-2, Cronbach’s α was 0.97 in females and 0.96 in males.

The Lithuanian version of the Drive for Muscularity Scale (DMS) was used. It reflects body image attitudes and behaviours towards body muscularity irrespective of a person’s actual muscle mass [53]. The response options range in a six-point Likert scale from 1 (never) to 6 (always). A total averaged score indicates higher drive for muscularity attitudes and behaviours. The psychometric properties of the Lithuanian translation of the DMS were previously tested in a sample of male students [54]. In this study, for the DMS, Cronbach α was 0.90 in females and 0.91 in males.

The Lithuanian version of the Sociocultural Attitudes towards Appearance Questionnaire 4 (SATAQ-4) was used. SATAQ-4 is one of the most widely used measures of sociocultural factors that contribute to an acceptance of prevailing appearance ideal [55]. The SATAQ-4 consists of five subscales: internalization of a thin/low-body-fat body beauty ideal, internalization of a muscular/athletic body beauty ideal and perceived pressure on appearance from the media, family, and peers. For this study, an additional subscale was created to reflect perceived pressure on appearance from a coach. The items of coach pressure subscale were: “I feel pressure from coach to look more thinner”; “I feel pressure from coach to enhance appearance”; “I feel pressure from coach to decrease body fat”; and “My coach encourages me to decrease body weight for getting better body shape”. All items are scored on a five-point Likert scale from 1 (definitely disagree) to 5 (definitely agree). The final scores are calculated by averaging the response options for each subscale and for the total scale. Previously, the psychometric properties and the original five-factor structure of the Lithuanian translation of the SATAQ-4 was confirmed in a student sample [56]. In this study, for the SATAQ-4, Cronbach’s α was in the range of 0.85–0.98 in females and 0.85–0.92 in male for the subscales and 0.93 and 0.92 for the total score, respectively.

The questions about unhealthy and healthy dietary habits were taken from the national survey “Health Behavior among Lithuanian Adult Population” [57] and used in previous studies [58,59]. Unhealthy dietary habits reflect unhealthy food (sweets, chips, fast food) consumption for snacking, overeating, eating in a rush or eating while working or reading, and eating late at night less than 2 h before going to bed. Healthy dietary habits reflect regular eating regimens, having breakfast and lunch, using more healthy cooking techniques, and choosing healthy snacks (fruits, berries, vegetables). All dietary habits were assessed using a five-option Likert-type scale with the response options from 1 (never) up to 5 (always). The final averaged scores of unhealthy and healthy dietary habits represent more frequent specific dietary patterns.

Study participants were asked to indicate their weighing frequency during the past month by providing the response options from “never” (0) up to “several times a day” [6]. Similar question was used in the previous study [60].

### 2.4. Statistical Analysis

Descriptive statistics were calculated and the distribution of continuous variables tested for normality in a preliminary analysis. A chi-square test was used to assess the associations between pairs of categorical variables. Next, an independent-samples *t*-test was run to compare body image and DE in the two age groups with the Cohen’s d calculated to represent the effect size. Effect sizes above 0.2 were considered small, and those equal to or above 0.5 were considered moderate [61]. Binary logistic regression was run to test independent multivariable adjusted effects of gender, age, and sensitivity in sports groups on any occurrence of the DE during the last week. 

Finally, analysis of covariance (ANCOVA) was employed to test the DE and BI differences across three weight-sensitive sports groups where age was added as a covariate. A *Bonferroni* post hoc test was used for multiple pairwise comparisons between groups. The effect sizes, represented by eta-squared, were calculated. An effect size above 0.01 and below 0.06 was considered small, above 0.06 and below 0.12 moderate, and ≥0.12 as large [61]. A *p*-value less than 0.05 was considered as statistically significant. All statistical analyses were carried out with SPSS v. 29 (IBM Corp., Armonk, NY, USA).

## 3. Results

### 3.1. The Analysis of the Effect of Age on Study Variables 

Table 1 presents a comparison of disordered eating (DE) behaviours during the last 28 days across age groups separately in males and females. Notably, dietary restraint behaviours were more prevalent in male athletes aged 18 years and older as compared with those <18 years. There were no significant differences when other types of DE were compared in adolescent male athletes and adults. Next, adolescent female athletes aged <18 years demonstrated more frequent self-induced vomiting, laxative misuse, and excessive exercise to reduce body weight during the last 28 days when compared with the age group ≥18 years. 

Multivariable adjusted effects of gender (male was the reference group), age (<18 years was the reference group), and weight sensitivity in the sports group (weight-sensitive aesthetic sports group was the reference) on the occurrence of any DE behaviour during the last 28 days are presented in Table 2 with the odds ratios obtained from binary logistic regression. The models were tested on each DE separately. Females were more likely to report dietary restraint in the last 28 days, binge eating, and excessive exercise to reduce body weight by 96%, 47%, and 51%, respectively, as compared with males. Participation in less WS sports reduced the odds of dietary restraint by 49% and excessive exercise by 43% as compared with the WS aesthetic sports group. Furthermore, athletes participating in other than aesthetic WS sports were less likely to engage in excessive exercise by 44% as compared with the aesthetic sports group. Finally, older age was associated with reduced odds of vomiting, laxative misuse, and excessive exercise by 49%, 56%, and 31%, respectively, as compared with the age group <18 years. 

In Table 3, the comparison of the study measures between female athletes <18 years and adult athletes ≥18 years is presented. It was revealed that adult athletes had higher BMI, greater body appreciation, and more healthy eating habits as compared with adolescents <18 years. On the other hand, adolescent athletes demonstrated more perceived pressures regarding appearance from family, peers, and coaches and had a higher total SATAQ-4 score. All Cohen’s d coefficients demonstrated a small-to-medium effect size.

In Table 4, the comparison of the study measures between adolescent male athletes <18 years and adult male athletes ≥18 years are presented. It was revealed that adult athletes had higher BMI, more frequent unhealthy dietary habits, and higher levels of DE and overweight preoccupation. Furthermore, adult male athletes indicated more frequent self-weighing. All Cohen’s d coefficients demonstrated a small-to-medium effect size.

### 3.2. Analysis of the Effect of Weight Sensitivity on the Study Variables 

Finally, DE and BI were compared across three groups of different levels of WS in sports separately in male and female athletes controlling for age (Table 5). Thus, the independent effects of weight sensitivity in sports on DE and BI could be explored. Female athletes participating in less WS sports had higher BMI and attained a higher score on DMS. By contrast, athletes involved in aesthetic WS sports perceived more pressure on appearance from their coaches, were more preoccupied with overweight, demonstrated more frequent self-weighing behaviour, and scored higher on the EDE-Q 6. Furthermore, in the aesthetic WS sports group, the score for healthy dietary habits was higher compared with the other groups.

The same differences across WS in sports groups in the male subsample were not apparent (Table 6). Athletes from less WS sports groups had higher BMI and attained a higher score for self-classified weight. By contrast, athletes participating in aesthetic WS sports demonstrated more frequent self-weighing behaviour than those from the less WS sports group.

Finally, in female athletes, the correlations between the EDE-Q 6 score and body appreciation were from −0.61 to −0.63 (*p* < 0.001) and similar across different WS in sports groups. The same correlations in male athletes varied from −0.15 in the less WS group, to −0.27 in the other WS group, to −0.32 in the aesthetic WS group (*p* < 0.05). Furthermore, the correlations between overweight preoccupation and the total EDE-Q 6 score were from 0.74 to 0.79 in different WS sports groups in female and from 0.56 to 0.60 in male athletes (*p* < 0.001).

## 4. Discussion

### 4.1. Effect of Age on DE, Body Image, and Sociocultural and Coach-Related Pressures

In the present study, we aimed to compare DE, body image, and general sociocultural and coach-related pressures in groups of competitive athletes of different age (adolescents and adults). We hypothesized that DE, body image concerns, and sociocultural pressures would be more prevalent in younger athletes. The present study adds important new knowledge that age is associated with DE, body image, and sociocultural pressures for competitive female athletes. Specifically, we observed that adolescent competitive female athletes reported more frequent self-induced vomiting, laxative misuse, and excessive exercise, as well as poorer heathy eating habits, compared with adult female athletes. Further, adolescent female athletes expressed a greater desire to attain stereotypical body ideals and higher pressures to attain a stereotyped body image from family, peers, and coaches, compared with adult female athletes. This finding is novel; therefore, the comparison of the results is limited, and future studies should test it. A recent study reported that parents of adolescent athletes might use psychological violence and neglect towards their children regarding body weight, especially in adolescent girls [62]. A plethora of studies reported that coach pressures on the body image of athletes is associated with their body image concerns and DE [63]. Peers, especially teammates, are also an important source of pressure regarding appearance and body weight in adolescent athletes [64]. Adult female athletes may no longer experience appearance and body-weight-related pressures from their parents. Further, adult athletes spent more years in sport, possibly facing pressures from the sport-related environment [14]. However, they have a more stable athletic identity, higher athletic achievements, and might also have higher resilience to sociocultural pressures, including pressures from coaches. Stronger athletic identity have been associated with enhanced athletic performance, improved global self-esteem and confidence, as well as improved social relationships [65,66]. Since the study is cross-sectional, it is impossible to understand the directions of the associations. It might be that younger athletes experience more intense general sociocultural and coach-related pressures on their appearance; however, it might also be that those young female athletes, who experience high general sociocultural and coach-related pressures, have higher body image concerns and DE and withdraw from sports more frequently compared to those who experience less pressure. Future studies with experimental and longitudinal designs should be implemented to understand this topic. 

Finally, the results of the present study showed that adult female athletes have a significantly more positive body image (body appreciation) compared with adolescents. Body appreciation is protective of body image in women and girls [35]. Higher body appreciation means that female athletes express respect and love for their bodies and have more resilience to the sociocultural pressures to attain stereotyped body ideals [67]. These findings suggest that special programs helping adolescent female competitive athletes to promote positive body image and resist sociocultural and sport-related pressures should be implemented. Special education for coaches who train adolescent females should be implemented to teach them how to avoid negative pressure on the body image of growing adolescent girls [40]. 

Importantly, no effect of age on the sociocultural pressures on body image was observed in male athletes. However, we found that higher dietary restraint, DE, self-weighing frequency, unhealthy nutrition habits and overweight preoccupation were more prevalent in adult male competitive athletes compared with adolescents. These results might be explained by the high level of competition and sport-related expectations that adult male athletes competing in elite sport possibly experience. Young adolescent athletes are in their puberty period, and it might be that they adjust body-weight-related expectations more easily compared with adult males. Since knowledge on this topic is very limited, these speculations should be tested by future studies. However, special efforts to prevent DE and body image concerns in adult male athletes are necessary. Furthermore, it is important to educate all adult male athletes about healthy nutrition and safe body weight control methods to prevent DE.

### 4.2. Effect of Weight Sensitivity in Sports on DE, Body Image, and Sociocultural and Coach-Related Pressures

Further, we hypothesized that DE, body image concerns, and sociocultural pressures on appearance would be more prevalent in athletes participating WS sports, controlling for age and gender. This hypothesis was partially confirmed. In females, DE was more prevalent in aesthetic sports compared with less WS sports; however, no significant differences were observed between less WS sports and other WS sports and between aesthetic WS sports and other WS sports, controlling for age. Further, we observed a significant effect of weight sensitivity for dietary restraint and excessive exercise. Specifically, we observed that female gender and participation in WS (aesthetic) sports predicted higher dietary restraint and excessive exercise in females, controlling for age. However, no significant effect of weight sensitivity in sport groups was observed for binge eating, vomiting and laxative misuse in females. These results are in accordance with previous studies which showed that disordered eating is more prevalent in female athletes participating in aesthetic sports compared with other athletes [4,6,23,68]. The present study extends previous knowledge that both adolescent and adult female athletes participating in aesthetic sports experience significantly higher DE than athletes participating in less WS sports. 

Our results also add important new knowledge that, despite having the lowest body weight, females participating in aesthetic sports report the highest overweight preoccupation, which is one of the main facets of negative body image [49] and DE [69]. These findings are in line with studies reporting that females participating in aesthetic sports report a more negative body image than other athletes [32]. Further, female adults involved in aesthetic sports reported a higher self-weighing frequency compared with other weight-sensitive groups. Frequent self-weighing is related to overweight preoccupation and considered a risk factor for DE in female athletes [60,70]. 

The present study adds important knowledge that aesthetic female athletes do not express a more positive body image (body appreciation) than females participating in other WS sports and less WS sports. It is an important novel finding suggesting that, despite low body weight, female athletes involved in aesthetic sport do not develop a healthier body image that might protect them from disordered eating. Further, we observed that body appreciation is negatively associated with disordered eating in all three groups of female athletes. This is a novel finding that is in line with the modern DE prevention recommendations that are based on cognitive dissonance principles and the promotion of positive body image and self-compassion in female athletes [2,71,72].

Finally, the present study adds new knowledge that aesthetic sports involving females report significantly higher pressure from coaches to attain appearance ideals compared with other female athletes. Previous studies showed that pressure from coaches is one of the main risk factors for DE in athletes [24,41]. These results clearly suggest that female athletes involved in aesthetic sports experience higher body image concerns and DE compared with competitive athletes participating in less WS sports. Positive body image and self-efficacy promotion programs for female athletes participating in WS sports are of immense importance. Special education (the prevention of DE and the promotion of positive body image) for coaches of WS sports, especially aesthetic sports, must be implemented [6,73,74,75]. 

In male competitive athletes, no significant differences were observed between WS and less WS groups in DE, body image, and sociocultural and coach-related pressures, controlling for age. These results contradict previous findings that showed a higher prevalence of DE in male athletes participating in aesthetic sports compared with other athletes [8]. This might be explained by an extremely low number of male aesthetic sports participants in the present sample, which might be considered as a limitation of the present study. Therefore, the generalization of these results should be limited. However, we observed a higher self-weighing frequency and self-classified weight in a group of male athletes participating in other WS sports compared with less WS sports. Previous studies reported that self-weighing is associated with an increased risk of eating disorders in male athletes [76]. These results suggest that males participating in sports where lower body weight is advantageous and those participating in sports with weight categories also need education about safe methods of weight loss and the risks of rapid weight loss, which are prevalent in WS sports with weight categories [16]. Finally, we observed that positive body image was negatively associated with DE in less WS and other than aesthetic WS sports groups in male athletes. Thus, positive body image promotion might also be effective in DE prevention programs for men.

### 4.3. Limitations and Strengths of the Study

The present study has important limitations. First, it is a cross-sectional study, and the direction of the associations is unclear. Specifically, the cross-sectional design limits the ability to establish causal relationships and determine the direction of associations. Experimental or longitudinal studies are recommended to have more clarity on the associations between study variables. Second, few male athletes from aesthetic sports participated in the present research. This might have an impact on the results of the comparison of study variables between groups of weight sensitivity in sports, especially comparing aesthetic sports and less weight-sensitive sports. Next, the point of the season and years in sport might influence the results of the study [14]. Another limitation of the present study is that the sample is convenient and does not represent any population. Furthermore, in the present study, we used a modified SATAQ-4 subscale to assess coach pressures on athletes’ body image. This might be considered as a limitation of the study. Future studies might additionally include other instruments that are widely accepted for assessing coach and sport-related pressures on athletes’ body image [77,78]. 

The strengths of the study include the large sample of competitive athletes from a variety of sports, including WS and less WS sports, of different genders and ages, and the use of sound research instruments to test DE [79], body image, and sociocultural pressures to attain a stereotyped body image. 

### 4.4. Practical Implications

Special DE prevention and positive body image promotion programs are necessary for female competitive athletes participating in weight-sensitive sports (especially aesthetic sports). Special education programs aiming to increase resistance to general sociocultural and coach-specific pressures of adolescent female athletes should be implemented. For adult male athletes participating in competitive sports, special programs that aim to prevent disordered eating and to increase healthy eating habits might be beneficial. Special education is essential for coaches who train adolescent and adult female athletes participating in weight-sensitive sports, especially aesthetic ones.

## 5. Conclusions

The results of the present study extend the existing knowledge about the effect of age and weight sensitivity on disordered eating in competitive athletes. When the effect of age was tested, higher DE behaviours (self-induced vomiting, laxative misuse, and excessive exercise) in adolescent female athletes were observed compared with adults. Adolescent female athletes experienced higher sociocultural pressures (family, peers) and sport-related pressures (coach) compared with adult female athletes. No differences in negative body image were observed; however, adult female athletes reported significantly higher positive body image (body appreciation) than adolescents. Adult male athletes more frequently experienced dietary restraint, disordered eating, unhealthy eating habits, self-weighing behaviour, and overweight preoccupation compared with adolescent male athletes; however, no differences were observed in sociocultural pressures and positive body image in the two age groups. 

In the present study, we observed a significantly higher prevalence of disordered eating, body image concerns, self-weighing practice, and body-image-related pressures from coaches in aesthetic-sport-involved female athletes compared with athletes participating in less weight-sensitive sports. No differences in positive body image (body appreciation) were observed between weight-sensitive and less weight-sensitive female sports participants. In male athletes, no significant differences were observed between weight-sensitive and less weight-sensitive sports in disordered eating and body-image-related sociocultural pressures. 

Special disordered eating prevention and positive body image promotion programs are necessary for female competitive athletes and parents of adolescent female athletes, especially those participating in aesthetic sports. For adult male athletes, special programs aiming to promote healthy eating should be implemented to prevent DE and body image concerns. Special education about DE prevention is essential for coaches who train female athletes. 

## Figures and Tables

**Table 1 nutrients-15-02724-t001:** The comparison of any occurrence (%) of disordered eating behaviours across age groups during the last 28 days in male and female athletes (*n* = 1003).

Disordered Eating Behaviors	Males <18 Years (*n* = 276)	Males ≥18 Years (*n* = 212)	χ^2^; *p*
Dietary restraint	17.0	24.5	4.2; 0.041
Binge eating (loss of control)	38.4	46.2	3.0; 0.083
Self-induced vomiting	10.1	6.1	2.5; 0.113
Laxative misuse	9.4	5.2	3.1; 0.08
Excessive exercise	51.1	46.2	1.1; 0.287
	Females < 18 years (*n* = 286)	Females ≥ 18 years (*n* = 229)	
Dietary restraint	36.4	31.0	1.6; 0.202
Binge eating (loss of control)	55.2	47.2	3.3; 0.068
Self-induced vomiting	7.7	3.5	4.1; 0.043
Laxative misuse	8.0	3.1	5.8; 0.016
Excessive exercise	67.1	52.8	10.9; 0.001

**Table 2 nutrients-15-02724-t002:** The effects of gender, age, and weight-sensitivity group on the odds ratios * of disordered eating behaviours during the last 28 days in the sample of athletes (*n* = 1003).

Dietary Restraint		B	OR	95% CI	*p*
Gender	Male	1.0			
	Female	0.68	1.96	1.46–2.64	<0.001
Age group	<18	1.0			
	≥18	0.11	1.12	0.84–1.49	0.448
WS group in sports	WS aesthetic	1.0			
	WS other	−0.29	0.75	0.49–1.14	0.174
	Less WS	−0.67	0.51	0.34–0.78	0.002
Binge eating					
Gender	Male	1.0			
	Female	0.38	1.47	1.13–1.89	0.003
Age group	<18	1.0			
	≥18	−0.02	0.98	0.76–1.26	0.869
WS group in sports	WS aesthetic	1.0			
	WS other	−0.09	0.91	0.61–1.36	0.648
	Less WS	0.02	1.03	0.69–1.51	0.902
Vomiting					
Gender	Male	1.0			
	Female	−0.39	0.67	0.41–1.12	0.126
Age group	<18	1.0			
	≥18	−0.67	0.51	0.30–0.87	0.012
WS group in sports	WS aesthetic	1.0			
	WS other	−0.05	0.96	0.43–2.13	0.910
	Less WS	0.07	1.08	0.49–2.35	0.853
Laxative misuse					
Gender	Male	1.0			
	Female	−0.32	0.73	0.43–1.22	0.231
Age group	<18	1.0			
	≥18	−0.81	0.44	0.25–0.78	0.004
WS group in sports	WS aesthetic	1.0			
	WS other	−0.29	0.75	0.34–1.66	0.474
	Less WS	−0.03	0.97	0.45–2.06	0.930
Excessive exercise					
Gender	Male	1.0			
	Female	0.41	1.51	1.17–1.96	0.002
Age group	<18	1.0			
	≥18	−0.38	0.69	0.53–0.89	0.004
WS group in sports	WS aesthetic	1.0			
	WS other	−0.59	0.56	0.36–0.85	0.007
	Less WS	−0.57	0.57	0.37–0.86	0.007

* Binary logistic regression; B = regression coefficient; OR = odds ratio; CI = confidence interval; WS = weight sensitive.

**Table 3 nutrients-15-02724-t003:** The comparison of the study measures (m ± SD) in female athletes across <18 and ≥18 years age groups (*n* = 515).

Study Measures	<18 Years (*n* = 286)	≥18 Years (*n* = 229)	Cohen’s d	*p*
Body mass index, kg/m^2^	20.37 ± 2.39	21.89 ± 2.35	−0.64	<0.001
EDE-Q 6	1.30 ± 1.13	1.12 ± 1.01	-	0.067
BAS-2	3.42 ± 1.07	3.76 ± 0.89	−0.38	<0.001
MBSRQ-AS: OP	2.42 ± 0.97	2.50 ± 0.98	-	0.363
MBSRQ-AS: SCW	3.04 ± 0.61	3.06 ± 0.52	-	0.731
SATAQ-4: total	2.59 ± 0.77	2.43 ± 0.73	0.22	0.015
SATAQ-4: thin	3.27 ± 1.17	3.09 ± 1.11	-	0.072
SATAQ-4: muscular	3.45 ± 0.91	3.34 ± 0.88	-	0.147
SATAQ-4: pressures media	2.54 ± 1.46	2.66 ± 1.50	-	0.361
SATAQ-4: pressures family	1.87 ± 1.01	1.64 ± 0.94	0.24	0.008
SATAQ-4: pressures peers	1.79 ± 1.12	1.60 ± 0.93	0.18	0.04
SATAQ-4: pressures coach	2.26 ± 1.21	1.88 ± 1.10	0.33	<0.001
DMS	2.36 ± 0.95	2.21 ± 0.95	-	0.075
Self-weighing frequency	2.95 ± 1.59	3.08 ± 1.68	-	0.392
Unhealthy nutrition habits	2.78 ± 0.58	2.82 ± 0.55	-	0.378
Healthy nutrition habits	3.51 ± 0.63	3.68 ± 0.57	−0.29	0.001

m = mean, SD = standard deviation; EDE-Q 6 = Eating Disorder Examination Questionnaire 6; MBSRQ-AS = Multidimensional Self-Relations Questionnaire, Appearance Scales; OP = Overweight Preoccupation; SCW = Self-Classified Weight; DMS = Drive for Muscularity Scale; SATAQ-4 = Sociocultural Attitudes towards Appearance Questionnaire 4; BAS-2 = Body Appreciation Scale 2.

**Table 4 nutrients-15-02724-t004:** Comparison of the study measures (m ± SD) in male athletes across <18 and ≥18 years age groups (*n* = 488).

Study Measures	<18 Years (*n* = 276)	≥18 Years (*n* = 212)	Cohen’s d	*p*
Body mass index, kg/m^2^	21.32 ± 3.03	23.37 ± 3.11	−0.67	<0.001
EDE-Q 6	0.51 ± 0.54	0.61 ± 0.62	−0.19	0.043
BAS-2	4.09 ± 0.82	4.05 ± 0.81	-	0.585
MBSRQ-AS: OP	1.90 ± 0.72	2.11 ± 0.82	−0.28	0.003
MBSRQ-AS: SCW	2.78 ± 0.56	2.80 ± 0.55	-	0.701
SATAQ-4: total	2.16 ± 0.66	2.10 ± 0.63	-	0.288
SATAQ-4: thin	2.30 ± 0.94	2.42 ± 0.91	-	0.157
SATAQ-4: muscular	3.32 ± 1.06	3.20 ± 0.96	-	0.213
SATAQ-4: pressures media	1.77 ± 1.03	1.69 ± 1.02	-	0.389
SATAQ-4: pressures family	1.71 ± 0.91	1.56 ± 0.81	-	0.067
SATAQ-4: pressures peers	1.64 ± 0.89	1.54 ± 0.85	-	0.207
SATAQ-4: pressures coach	1.90 ± 0.98	1.81 ± 0.95	-	0.331
DMS	3.01 ± 1.12	2.83 ± 1.12	-	0.083
Self-weighing frequency	3.16 ± 1.62	3.55 ± 1.61	−0.24	0.009
Unhealthy nutrition habits	2.63 ± 0.63	2.76 ± 0.58	−0.23	0.014
Healthy nutrition habits	3.65 ± 0.59	3.65 ± 0.56	-	0.993

m = mean, SD = standard deviation; EDE-Q 6 = Eating Disorder Examination Questionnaire 6; MBSRQ-AS = Multidimensional Self-Relations Questionnaire, Appearance Scales; OP = Overweight Preoccupation; SCW = Self-Classified Weight; DMS = Drive for Muscularity Scale; SATAQ-4 = Sociocultural Attitudes towards Appearance Questionnaire 4; BAS-2 = Body Appreciation Scale 2.

**Table 5 nutrients-15-02724-t005:** Comparison of the study measures m * (95% CI) across sports groups of different weight sensitivities in female athletes (*n* = 515).

Study Measures	Aesthetic Weight-Sensitive Sports *n* = 105	Other Weight-Sensitive Sports *n* = 165	Less Weight-Sensitive Sports *n* = 245	Eta-Squared; *p*
Body mass index, kg/m^2^	19.68 (19.25–20.11)	20.93 ^a^ (20.58–21.27)	21.71 ^ab^ (21.43–22.00)	0.11; <0.001
EDE-Q 6	1.47 (1.26–1.68)	1.23 (1.07–1.40)	1.10 ^a^ (0.97–1.24)	0.02; 0.017
BAS-2	3.58 (3.39–3.78)	3.56 (3.41–3.71)	3.57 (3.44–3.70)	0.983
MBSRQ-AS: OP	2.74 (2.55–2.93)	2.41 ^a^ (2.26–2.56)	2.37 ^a^ (2.24–2.49)	0.02; 0.004
MBSRQ-AS: SCW	2.94 (2.83–3.05)	3.05 (2.96–3.13)	3.10 (3.03–3.18)	0.058
SATAQ-4: total	2.60 (2.46–2.75)	2.59 (2.48–2.71)	2.44 (2.35–2.54)	0.069
SATAQ-4: thin	3.38 (3.16–3.60)	3.25 (3.07–3.42)	3.07 (2.92–3.21)	0.051
SATAQ-4: muscular	3.34 (3.17–3.51)	3.50 (3.36–3.64)	3.36 (3.25–3.47)	0.219
SATAQ-4: pressures media	2.60 (2.31–2.89)	2.72 (2.49–2.95)	2.51 (2.32–2.70)	0.373
SATAQ-4: pressures family	1.67 (1.48–1.86)	1.78 (1.63–1.93)	1.79 (1.67–1.92)	0.578
SATAQ-4: pressures peers	1.65 (1.45–1.85)	1.72 (1.56–1.88)	1.73 (1.60–1.86)	0.789
SATAQ-4: pressures coach	2.60 (2.38–2.82)	2.19 ^a^ (2.02–2.36)	1.80 ^ab^ (1.66–1.94)	0.07; <0.001
Self-weighing frequency	3.57 (3.26–3.88)	3.04 ^a^ (2.79–3.29)	2.75 ^a^ (2.55–2.95)	0.04; <0.001
DMS	1.95 (1.77–2.14)	2.37 ^a^ (2.23–2.51)	2.38 ^a^ (2.26–2.50)	0.03; <0.001
Unhealthy nutrition habits	2.78 (2.67–2.89)	2.82 (2.73–2.90)	2.79 (2.72–2.86)	0.847
Healthy nutrition habits	3.67 (3.56–3.79)	3.70 (3.61–3.79)	3.47 ^ab^ (3.40–3.54)	0.03; <0.001

* = controlled by age; m = mean; CI = confidence interval. EDE-Q 6 = Eating Disorder Examination Questionnaire 6; MBSRQ-AS = Multidimensional Self-Relations Questionnaire, Appearance Scales; OP = Overweight Preoccupation; SCW = Self-Classified Weight; DMS = Drive for Muscularity Scale; SATAQ-4 = Sociocultural Attitudes towards Appearance Questionnaire 4; BAS-2 = Body Appreciation Scale 2. Etas squared are calculated only in case of significant differences between groups; ^a^ = *p* < 0.05 as compared with aesthetic weight-sensitive sports group; ^b^ = *p* < 0.05 as compared with the other weight-sensitive sports group.

**Table 6 nutrients-15-02724-t006:** Comparison of the study measures m * (95% CI) across sports groups of different weight sensitivities in male athletes (*n* = 488).

Study Measures	Aesthetic Weight-Sensitive Sports *n* = 31	Other Weight-Sensitive Sports *n* = 235	Less Weight-Sensitive Sports *n* = 222	Eta-Squared; *p*
Body mass index, kg/m^2^	20.69 (19.64–21.74)	22.19 ^a^ (21.81–22.57)	22.45 ^a^ (22.05–22.84)	0.02; 0.009
EDE-Q 6	0.68 (0.48–0.88)	0.56 (0.48–0.63)	0.53 (0.46–0.61)	0.403
BAS-2	3.89 (3.60–4.18)	4.11 (4.00–4.21)	4.05 (3.95–4.16)	0.372
MBSRQ-AS: OP	2.03 (1.76–2.30)	1.99 (1.90–2.09)	1.97 (1.87–2.07)	0.913
MBSRQ-AS: SCW	2.51 (2.32–2.70)	2.83 ^a^ (2.76–2.90)	2.77 ^a^ (2.70–2.85)	0.02; 0.009
SATAQ-4: total	2.28 (2.05–2.51)	2.10 (2.02–2.18)	2.15 (2.06–2.24)	0.304
SATAQ-4: thin	2.38 (2.05–2.71)	2.38 (2.26–2.50)	2.32 (2.19–2.44)	0.755
SATAQ-4: muscular	3.41 (3.05–3.77)	3.20 (3.07–3.33)	3.33 (3.19–3.46)	0.303
SATAQ-4: pressures media	2.00 (1.63–2.36)	1.72 (1.58–1.85)	1.73 (1.59–1.86)	0.351
SATAQ-4: pressures family	1.75 (1.44–2.06)	1.62 (1.50–1.73)	1.66 (1.54–1.77)	0.680
SATAQ-4: pressures peers	1.63 (1.32–1.94)	1.55 (1.44–1.67)	1.64 (1.53–1.76)	0.534
SATAQ-4: pressures coach	2.20 (1.86–2.54)	1.79 (1.67–1.91)	1.89 (1.76–2.02)	0.069
Self-weighing frequency	3.56 (3.00–4.12)	3.70 (3.50–3.90)	2.90 ^b^ (2.69–3.11)	0.06; <0.001
DMS	3.11 (2.72–3.51)	2.87 (2.73–3.01)	2.98 (2.83–3.12)	0.389
Unhealthy nutrition habits	2.60 (2.39–2.82)	2.71 (2.63–2.79)	2.67 (2.59–2.75)	0.599
Healthy nutrition habits	3.60 (3.39–3.80)	3.70 (3.62–3.77)	3.61 (3.53–3.68)	0.217

* = controlled by age; m = mean; CI = confidence interval. EDE-Q 6 = Eating Disorder Examination Questionnaire 6; MBSRQ-AS = Multidimensional Self-Relations Questionnaire, Appearance Scales; OP = Overweight Preoccupation; SCW = Self-Classified Weight; DMS = Drive for Muscularity Scale; SATAQ-4 = Sociocultural Attitudes towards Appearance Questionnaire 4; BAS-2 = Body Appreciation Scale 2. Etas squared are calculated only in case of significant differences between groups; ^a^ = *p* < 0.05 as compared with the aesthetic weight-sensitive sports group; ^b^ = *p* < 0.05 as compared with the other weight-sensitive sports group.

## Data Availability

The dataset generated and analysed during the current study is available from the corresponding author on reasonable request.

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
