# Peer review of "The Comparison of Disordered Eating, Body Image, Sociocultural and Coach-Related Pressures in Athletes across Age Groups and Groups of Different Weight Sensitivity in Sports"

_nutrients, 2023, doi:10.3390/nu15122724_

Round 1
Reviewer 1 Report
ABSTRACT AND KEYWORDS:
- In the list of keywords, there are some terms that are repetitive. For example: "body image" and "body appreciation", or "competitive sport" and "weight-sensitive sports". By choosing one of each pair, the keywords could be narrowed down to four that spanned different fields.
INTRODUCTION:
- Excessive use of citation #2 has been observed in the first three paragraphs of the Introduction. It is recommended to use more diverse bibliographic sources for this content.
- The content of the Introduction is very correct. The context is widely developed.
MATERIALS AND METHODS:
- The study design, the subjects description, the inclusion criteria and the exclusion criteria should appear clearly separated in subsections.
- Considering the manuscript topic, the BMI (which was also not measured directly, but rather by what the participants reported) is not adequate to assess the weight categories. It should have been through body composition using bioimpedance or anthropometry. Or, at least, in the case of BMI, that height and weight would have been measured.
- Line 205: What is considered an "exercise"?
- p significance value is not indicated in "Statistical analysis".
RESULTS:
- A wider Table 1 could have been made but with fewer rows (putting both males and females in columns), since the disordered eating behaviors are repeated. The same occurs in Table 2.
- The tables do not show significant differences (*).
Author Response
Dear Reviewer,
Thank you for your time reviewing our paper and for your comments. All changes made in the text are highlighted in a blue font.
ABSTRACT AND KEYWORDS:
- In the list of keywords, there are some terms that are repetitive. For example: "body image" and "body appreciation", or "competitive sport" and "weight-sensitive sports". By choosing one of each pair, the keywords could be narrowed down to four that spanned different fields.
The list of keywords was revised and shortened.
INTRODUCTION:
- Excessive use of citation #2 has been observed in the first three paragraphs of the Introduction. It is recommended to use more diverse bibliographic sources for this content.
Thank you for this comment. We included more diverse bibliographic sources in the first three paragraphs.
- The content of the Introduction is very correct. The context is widely developed.
Thank you for this comment.
MATERIALS AND METHODS:
- The study design, the subjects description, the inclusion criteria and the exclusion criteria should appear clearly separated in subsections.
We separated the subjects‘ descriptions, developed a section about inclusion and exclusion criteria, and reported the study design in the Methods section.
- Considering the manuscript topic, the BMI (which was also not measured directly, but rather by what the participants reported) is not adequate to assess the weight categories. It should have been through body composition using bioimpedance or anthropometry. Or, at least, in the case of BMI, that height and weight would have been measured.
We did not assess „weight categories“. We agree with the reviewer that objectively measured BMI is more accurate than self-reported, however, in the present study, BMI was never used as the dependent or independent variable. We assessed BMI only for sample characteristic purposes. Weight sensitivity was classified according to the classification proposed by other scholars and taking into account the sports type the athletes were engaged in.
- Line 205: What is considered an "exercise"?
By reporting hours of „exercise“ we mean training or workouts time.
- p significance value is not indicated in "Statistical analysis".
The statement about the p-value was included in the Statistical analysis.
RESULTS:
- A wider Table 1 could have been made but with fewer rows (putting both males and females in columns), since the disordered eating behaviours are repeated. The same occurs in Table 2.
- The tables do not show significant differences (*).
We decided not to revise tables for the clarity of reading. Table 1 aims to compare the prevalence of disordered eating behaviours across age groups separately in male and female athletes, while Table 2 presents data for the total sample aiming to test the effects of gender, age and sensitivity in sports group (independent variables) on disordered eating behaviours (dependent variables). A p-value is provided in the last column of each table. Providing (*) would be excessive in Tables 1-4, while for the Post Hoc comparisons in Tables 5 and 6, we used characters (ab).

Reviewer 2 Report
I have thoroughly reviewed the manuscript titled "Age and Weight Sensitivity Effects on Disordered Eating and Body Image in Competitive Athletes" submitted to Nutrients. The study provides valuable insights into the impact of age and weight sensitivity on disordered eating (DE) and body image in competitive athletes. Overall, the manuscript is well-written and presents a comprehensive analysis of the topic. The authors have addressed an important research gap and their findings have implications for athlete well-being and future research. However, there are a few areas that require attention and revision before publication.
Major Comments:
-
Clarity and Organization: The manuscript generally flows well, but certain sections could benefit from improved clarity and organization. The introduction provides a clear background and rationale for the study, while the summary of findings accurately presents the main results. However, it would be helpful to include a brief section outlining the methodology employed, including participant recruitment, measures used, and statistical analysis techniques. This will enhance the manuscript's overall clarity and allow readers to better understand the study design.
-
Discussion and Interpretation of Findings: The authors effectively discuss the findings in relation to existing literature, highlighting the unique contribution of their study. However, I encourage the authors to further elaborate on the potential reasons behind the observed differences in disordered eating and body image between adolescent and adult athletes. Are there specific developmental, social, or environmental factors that may explain these variations? Providing additional insights in this regard would strengthen the discussion and help readers grasp the underlying mechanisms at play.
-
Limitations and Future Directions: While the authors acknowledge some limitations of their study, it would be beneficial to expand upon them further. Specifically, the cross-sectional design limits the ability to establish causal relationships and determine the direction of associations. It would be valuable to discuss the potential impact of confounding variables and the need for longitudinal or experimental research to address these limitations. Additionally, the low representation of male athletes in aesthetic sports is a notable limitation that should be acknowledged and discussed in greater detail.
Minor Comments:
-
Language and Clarity: There are a few instances where the language could be refined for clarity and precision. Please carefully review the manuscript for grammatical errors, sentence structure, and ensure that concepts are explained in a concise and accessible manner. Additionally, consider providing definitions or clarifications for specialized terms or abbreviations to aid readers who may not be familiar with the field.
-
Tables and Figures: The inclusion of tables and figures greatly enhances the presentation of data. However, I recommend revising and proofreading the tables for consistency and clarity. Ensure that all abbreviations, symbols, and units of measurement are properly defined and consistently used throughout the manuscript.
-
Conclusion: The conclusion effectively summarizes the main findings and highlights the implications of the study. I suggest incorporating a brief section on the practical implications for athletes, coaches, and sports organizations. This will emphasize the relevance of the research and provide actionable takeaways for stakeholders in the field.
I suggest the following references to improve your article:
1) https://link.springer.com/article/10.1007/s40292-019-00352-2 --> wrong perception of parents about their child's weight
2) https://www.mdpi.com/2036-7503/14/4/49 --> how child's weight and eating habits changed with the COVID-19 pandemic
None to declare
Author Response
Dear Reviewer,
Thank you for your time reviewing our paper and for your comments. All changes made in the text are highlighted in a blue font.
I have thoroughly reviewed the manuscript titled "Age and Weight Sensitivity Effects on Disordered Eating and Body Image in Competitive Athletes" submitted to Nutrients. The study provides valuable insights into the impact of age and weight sensitivity on disordered eating (DE) and body image in competitive athletes. Overall, the manuscript is well-written and presents a comprehensive analysis of the topic. The authors have addressed an important research gap and their findings have implications for athlete well-being and future research. However, there are a few areas that require attention and revision before publication.
Major Comments:
- Clarity and Organization: The manuscript generally flows well, but certain sections could benefit from improved clarity and organization. The introduction provides a clear background and rationale for the study, while the summary of findings accurately presents the main results. However, it would be helpful to include a brief section outlining the methodology employed, including participant recruitment, measures used, and statistical analysis techniques. This will enhance the manuscript's overall clarity and allow readers to better understand the study design.
Section „Methods” presenting participants‘ recruitment, measures used, and statistical analysis is included in the manuscript.
- Discussion and Interpretation of Findings: The authors effectively discuss the findings in relation to existing literature, highlighting the unique contribution of their study. However, I encourage the authors to further elaborate on the potential reasons behind the observed differences in disordered eating and body image between adolescent and adult athletes. Are there specific developmental, social, or environmental factors that may explain these variations? Providing additional insights in this regard would strengthen the discussion and help readers grasp the underlying mechanisms at play.
Thank you for this remark. We included explanations about possible factors that might explain differences in body image and DE in female athletes.
- Limitations and Future Directions: While the authors acknowledge some limitations of their study, it would be beneficial to expand upon them further. Specifically, the cross-sectional design limits the ability to establish causal relationships and determine the direction of associations. It would be valuable to discuss the potential impact of confounding variables and the need for longitudinal or experimental research to address these limitations. Additionally, the low representation of male athletes in aesthetic sports is a notable limitation that should be acknowledged and discussed in greater detail.
Thank you for this comment, we included additional arguments in the study‘s limitation section.
Minor Comments:
- Language and Clarity: There are a few instances where the language could be refined for clarity and precision. Please carefully review the manuscript for grammatical errors, sentence structure, and ensure that concepts are explained in a concise and accessible manner. Additionally, consider providing definitions or clarifications for specialized terms or abbreviations to aid readers who may not be familiar with the field.
The text was double-checked.
- Tables and Figures: The inclusion of tables and figures greatly enhances the presentation of data. However, I recommend revising and proofreading the tables for consistency and clarity. Ensure that all abbreviations, symbols, and units of measurement are properly defined and consistently used throughout the manuscript.
The tables were double–checked for clarity.
- Conclusion: The conclusion effectively summarizes the main findings and highlights the implications of the study. I suggest incorporating a brief section on the practical implications for athletes, coaches, and sports organizations. This will emphasize the relevance of the research and provide actionable takeaways for stakeholders in the field.
The subsection Practical Implications was included.
I suggest the following references to improve your article:
1) https://link.springer.com/article/10.1007/s40292-019-00352-2 --> wrong perception of parents about their child's weight
2) https://www.mdpi.com/2036-7503/14/4/49 --> how child's weight and eating habits changed with the COVID-19 pandemic
Thank you, these articles are important, yet not related to the topic of the manuscript.
